# An Implantable Magneto-Responsive Poly(aspartamide) Based Electrospun Scaffold for Hyperthermia Treatment

**DOI:** 10.3390/nano12091476

**Published:** 2022-04-26

**Authors:** Tamás Veres, Constantinos Voniatis, Kristóf Molnár, Dániel Nesztor, Daniella Fehér, Andrea Ferencz, Iván Gresits, György Thuróczy, Bence Gábor Márkus, Ferenc Simon, Norbert Marcell Nemes, Mar García-Hernández, Lilla Reiniger, Ildikó Horváth, Domokos Máthé, Krisztián Szigeti, Etelka Tombácz, Angela Jedlovszky-Hajdu

**Affiliations:** 1Laboratory of Nanochemistry, Department of Biophysics and Radiation Biology, Semmelweis University, 1089 Budapest, Hungary; veres454@gmail.com (T.V.); constantinosvoniatis@gmail.com (C.V.); molnar.182@osu.edu (K.M.); 2Department of Surgery, Transplantation and Gastroenterology, Semmelweis University, 1082 Budapest, Hungary; 3Department of Food Engineering, University of Szeged, 6725 Szeged, Hungary; nesztor@chem.u-szeged.hu (D.N.); e.tombacz@chem.u-szeged.hu (E.T.); 4Heart and Vascular Centre, Department of Surgical Research and Techniques, Semmelweis University, 1122 Budapest, Hungary; daniella.feher@gmail.com (D.F.); ferencz.andrea@med.semmelweis-univ.hu (A.F.); 5Department of Biophysics and Radiation Biology, Semmelweis University, 1094 Budapest, Hungary; gresits.ivan@gmail.com (I.G.); horvath.ildiko@med.semmelweis-univ.hu (I.H.); mathe.domokos@med.semmelweis-univ.hu (D.M.); krisztian.szigeti@gmail.com (K.S.); 6NRIRR “Frédéric Joliot-Curie” National Research Institute for Radiobiology and Radiohygiene, 1221 Budapest, Hungary; thuroczy@hp.osski.hu; 7Stavropoulos Center for Complex Quantum Matter, Department of Physics and Astronomy, University of Notre Dame, Notre Dame, IN 46556, USA; bmarkus@nd.edu; 8Institute of Physics, Budapest University of Technology and Economics, 1521 Budapest, Hungary; simon.ferenc@ttk.bme.hu; 9Wigner Research Centre for Physics Economics, 1121 Budapest, Hungary; 10Grupo de Física de Materiales Complejos (GFMC), Departamento de Física de Materiales, Universidad Complutense de Madrid, 28040 Madrid, Spain; nmnemes@fis.ucm.es (N.M.N.); marmar@icmm.csic.es (M.G.-H.); 11Department of Pathology and Experimental Cancer Research, Semmelweis University, 1085 Budapest, Hungary; reiniger.lilla@med.semmelweis-univ.hu; 12Hungarian Center of Excellence for Molecular Medicine (HCEMM), In Vivo Imaging Advanced Core Facility, Semmelweis University Site, 1094 Budapest, Hungary; 13Soós Ernő Water Technology Research and Development Center, University of Pannonia, 8800 Nagykanizsa, Hungary

**Keywords:** theranostics, magnetic iron oxide nanoparticles, polysuccinimide, electrospinning, hyperthermia, MRI

## Abstract

When exposed to an alternating magnetic field, superparamagnetic nanoparticles can elicit the required hyperthermic effect while also being excellent magnetic resonance imaging (MRI) contrast agents. Their main drawback is that they diffuse out of the area of interest in one or two days, thus preventing a continuous application during the typical several-cycle multi-week treatment. To solve this issue, our aim was to synthesise an implantable, biodegradable membrane infused with magnetite that enabled long-term treatment while having adequate MRI contrast and hyperthermic capabilities. To immobilise the nanoparticles inside the scaffold, they were synthesised inside hydrogel fibres. First, polysuccinimide (PSI) fibres were produced by electrospinning and crosslinked, and then, magnetitc iron oxide nanoparticles (MIONs) were synthesised inside and in-between the fibres of the hydrogel membranes with the well-known co-precipitation method. The attenuated total reflectance Fourier-transform infrared spectroscopy (ATR-FTIR) investigation proved the success of the chemical synthesis and the presence of iron oxide, and the superconducting quantum interference device (SQUID) study revealed their superparamagnetic property. The magnetic hyperthermia efficiency of the samples was significant. The given alternating current (AC) magnetic field could induce a temperature rise of 5 °C (from 37 °C to 42 °C) in less than 2 min even for five quick heat-cool cycles or for five consecutive days without considerable heat generation loss in the samples. Short-term (1 day and 7 day) biocompatibility, biodegradability and MRI contrast capability were investigated in vivo on Wistar rats. The results showed excellent MRI contrast and minimal acute inflammation.

## 1. Introduction

Nanotechnology is an intensively developing field [1,2], making an impactful presence in both basic research and industry [3,4]. Amongst many possible applications, nanoparticles have the potential to be utilised as theranostics agents [5].

Theranostic agents (used in theranostics) concurrently combine both therapeutic and diagnostic features, a feat particularly advantageous in cancer management. These ensure that no undesirable differences occur between the biodistribution of diagnostic and therapeutic agents. The long-term goal of using theranostic agents is to monitor the effects of treatments while fine-tuning the therapy according to the specific needs of the patient, thus achieving personalised medicine [6]. 

Magnetic iron oxide nanoparticles (MIONPs) are highly promising candidates for theranostics [7,8,9,10,11,12]. Being superparamagnetic nanoparticles [13,14] their magnetic momentum turns towards an external magnetic field and thus act as nanomagnets. However, as soon as the external magnetic field ceases, they lose their magnetic momentum (or macroscopic momentum). These properties not only make them excellent magnetic resonance imaging (MRI) contrast agents but, more importantly, give them the ability to produce a magnetic hyperthermic effect which could play a key role in the future of cancer management.

Magnetic hyperthermia is still a promising complementary treatment (its in preclinical phase) for cancers, besides chemotherapy and radiotherapy, aiming to reduce the amount of medications and radiation needed and consequently decreasing their side effects as well [15]. During a hyperthermic treatment, the vicinity of cancer cells is heated (to 40–45 °C), inducing cell death (cell apoptosis). Cancer cells are more sensitive to heat shock than healthy cells, making the procedure somewhat selective. While this effect can be achieved by utilising a whole-body, regional or local hyperthermia, the more localised the treatment is, the more concentrated the effect gets, and the smaller are the chances of complications (e.g., systemic side effects). For this exact reason, the medium via which the nanoparticles are transported and applied is crucial. Intravenously administrated and hydrogel-based systems are prone to poorly localised distribution and nanoparticle diffusion, making treatments less effective. Therefore, the need for a system that can be locally implanted, fixed and repeatedly activated for extended periods of time is grave.

In this regard, electrospun fibrous scaffolds could provide not only a solution, but also additional features and functions [16]. Electrospinning creates randomly oriented micro- or nanofibres, which are exploited in various medical applications [13,17]. Due to their large surface area-to-volume ratio, they can effectively entrap large quantities of liquid, small molecules and nanoparticles [13,18] which in the case of magnetic nanoparticles (MNPs) could provide better nanoparticle distribution and a more uniform hyperthermic effect.

Issues may arise due to a continuously induced cell death causing tissue defects. However, electrospun fibrous scaffolds could also circumvent this issue. Electrospun membranes can be designed to replicate the structure of the local extracellular matrix, resulting in a perfect template for cell adhesion, proliferation and differentiation. Therefore, during the post-operative regeneration process when potential defects are present due to cell death, this nanofibrous system could provide a template for healthy cell regeneration. The most critical parameter, however, regarding nanofibrous mesh fabrication is the utilised polymer [19].

Polysuccinimide (PSI) is a cyto- and biocompatible polymer. It hydrolyses in vivo to poly(aspartic acid) (PASP), which can then be eliminated via renal secretion and bowel excretion [7,8,14]. Thus, after a potential implantation, not only nanoparticles, but also the implant, will be eliminated [9]. Electrospun PSI systems have been documented and their physico-chemical properties are more than suitable for implantation, while their biodegradation period can be adjusted days, weeks or months [7].

MIONP systems have been investigated in the past. Zhang et al. monitored fibroblast cell growth on electrospun polycaprolactone—poly(ethylene glycol)-polycaprolacton PCL-PEG-PCL-magnetite (Fe_3_O_4_) membranes and showed that increasing the iron oxide content of the scaffold also increases cell viability [10]. Similar results were achieved by Wang et al., using electrospun poly-L-lactide membranes loaded with oleic acid-coated magnetite. Their results confirm that the presence of iron oxide enables better cell proliferation rates and improved adhesion [11]. Since the positive effect of MIONPs on cell adhesion and proliferation are well-known, we focused on the possible theranostic applications of electrospun scaffolds containing iron oxide.

Our objective was to fabricate PSI-based electrospun meshes loaded with MIONPs, which could be utilised for magnetic hyperthermia treatment. Physico-chemical characterisation, hyperthermic effect evaluation as well as MRI contrast assessment and biocompatibility studies were performed.

## 2. Materials and Methods

### 2.1. Materials

The used materials are listed here: L-aspartic acid (Sigma-Aldrich, Saint Louis, MO, USA), dimethylformamide (DMF) (VWR International, Leuven, Belgium), o-phosphoric acid (VWR International, Leuven, Belgium), 1,4-diaminobutane (DAB) (99%; Sigma-Aldrich, Saint Louis, MO, USA), imidazole (ACS reagent, ≥99%; Sigma-Aldrich, Saint Louis, MO, USA), citric acid monohidrate (ACS reagent, ≥99.9%; VWR, International, Leuven, Belgium), sodium chloride (99–100.5%; Sigma-Aldrich, Saint Louis, MO, USA), phosphate-buffered saline (PBS) (Tablet; Sigma-Aldrich, Saint Louis, USA), sodium hydroxide (VWR International, Leuven, Belgium), 5,5′-ditio-bis-(2-nitrobenzoic acid) (≥99%; Sigma-Aldrich, Saint Louis, MO, USA), ethylenediaminotetra acetic acid (≥99%; Sigma-Aldrich, Saint Louis, MO, USA), hydrochloric acid (37%; Saint Louis, USA), ammonium iron (II) sulphate hexahydrate (VWR International, Leuven, Belgium), iron (II) chloride tetrahydrate (VWR International, Leuven, Belgium), iron (III) chloride (97%; Sigma-Aldrich, Saint Louis, USA), ethanol (anhydrous, denatured; Reanal Labor, Budapest, Hungary), potassium permanganate (Reanal, Budapest, Hungary), hydroxylamine hydrochloride (ACS reagent, Acros Organics, Sigma-Aldrich, Saint Louis, MO, USA), ammonium acetate (ultrapure (UP); VWR International, Leuven, Belgium), acetic acid (glacial, 100%; Suprapur, Sigma-Aldrich, Saint Louis, USA), 1,10-phenanthroline (99%; Alfa Aesar, Sigma-Aldrich, Saint Louis, MO, USA) and chlorine dioxide (Solvocid, Budapest, Hungary). All the chemicals were of analytical grade and used as received. For aqueous solutions, UP water (Human Corporation ZeneerPower I Water Purification System, Seoul, South Korea) was used.

### 2.2. Synthesis and the Electrospinning of PSI

PSI was synthesised according to our previous works [13,14]. The main steps were shown as following: A mixture of L-aspartic acid and phosphoric acid with a 1:1 weight ratio was transferred to a container and then mixed by a rotary vacuum evaporator system (RV10 digital rotary evaporator; IKA, Staufen, Germany) at a 3 mbar pressure for 8 h at 180 °C (Appendix A). The resulting polymer was dissolved in dimethylformamide, then precipitated and washed with water and dried at room temperature. The final PSI average molecular weight was calculated to be 28,500 ± 3000 g/mol based on viscometry and the Kuhn–Mark–Houwink equation [13].

The electrospinning apparatus consisted of a Genvolt 73030P power supply (Shropshire, UK), a KD Scientific KDS100 pump (Holliston, MA, USA), a syringe and a needle. The collector was a static aluminium-coated flat wooden rectangle. The needle–collector distance was kept at 15 cm, the flow rate was 1 mL/hour, the polymer concentration was 25 *w*/*w*% PSI in DMF, and the voltage was 13.5 kV.

### 2.3. Preparation of the Magnetic Scaffold—Crosslinking and Synthesis of MIONPs Inside the Polymer Scaffold

To create crosslinks between the chains inside the fibres, PSI samples (d = 16 mm) were submerged in a 0.5 M DAB/EtOH solution (2.52 cm^3^ DAB filled with EtOH up to 50 cm^3^ in a volumetric flask) (Appendix A). Subsequently, to synthesize magnetic irod oxide inside the fibres scaffold, membranes were first submerged in a Fe(II)–Fe(III)–chloride solution (14.92 g FeCl_3_ and 12.6 g FeCl2·4H2O dissolved in 30 mL UP water) and then in a 3 M NaOH solution (2.37 g solid NaOH dissolved in 25 mL UP water). In the study, we focused on samples synthesised with DAB/EtOH, Fe(II)–Fe(III)–chloride and NaOH for 1 h, 30 min and 30 min, respectively (Appendix A). The optimisation of the treatment duration for these 3 solutions was discussed in the Appendix A. Samples were stored in UP or PBS (pH = 7.4, I = 150 mM) at room temperature and were used within 2 days. Samples synthesised in this way were designated as PSI-DAB-Magn.

### 2.4. Attenuated Total Reflectance Fourier-Transform Infrared Spectroscopy (ATR-FTIR)

For the FTIR studies, a JASCO 4700A device equipped with an ATR diamond crystal (JASCO Ltd., Budapest, Hungary, ATR Pro ONE) and with a DTGS detector was used. For taking the spectra, 128 parallel measurements were made in the 4000–400 cm^−1^ range, with a 4 cm^−1^ resolution in all cases. Final spectra were obtained after water, CO_2_ and baseline corrections. In addition, the FTIR spectra of non-treated PSI fibres, iron oxide and PSI-DAB-Magn samples were taken.

### 2.5. SEM, TEM and Dynamic Light Scattering (DLS)

For the SEM studies, samples were prepared in the following manner: PSI-based membranes were washed thoroughly with UP water for 1–2 days and then freeze-dried. A small part of the membranes was cut out and placed on a conductive tape with a tweezer for coating. Samples were then sputter-coated with palladium in 20–30 nm thickness with a 2SPI Sputter Coating System (SPI supplies, West Chester, PA, USA). Micrographs were taken using a ZEISS EVO 40 XVP scanning electron microscope equipped with an Oxford INCA X-ray spectrometer (EVISA). An accelerating voltage of 20 kV was applied. Fibre diameter was determined using ImageJ. In every case, 100 individual fibres were measured. Averages were provided with a standard error at a 0.95 confidence level, assuming a normal distribution.

TEM studies were performed on a soaking solution of the PSI-DAB-Magn. After washing out some magnetic particles from the membrane (with UP water), particles were dropped on a Cu grid in a highly diluted way and dried on it. The grid was placed in an FEI Morgagni transmission electron microscope (FEI company, Oregon USA), with a maximum 100 KV accelerating voltage. The maximum resolution of the instrument was 0.2 nm.

DLS measurements were carried out with an Anton Paar Litesizer 500 instrument (Anton Paar, Budapest, Hungary). For the measurements, diluted samples (with ultrafine water) were used in disposable cuvette. Three parallel measurements were carried out (each one kept for 60 s) in the back scattering mode (25 °C).

### 2.6. Superconducting Quantum Interference Device (SQUID)

The SQUID is a very sensitive magnetometer that is suitable to study the magnetic properties of a sample. We used a commercial (Quantum Design MPMS3–7T) SQUID for the measurements (Quantum Design GmbH, Darmstadt, Germany). Magnetic moment versus magnetic field measurements for consecutive loops can assess whether hysteresis or a finite coercive field is present in the samples. In principle, the absence of coercivity attests that the samples are superparamagnetic [20]. The saturation magnetic moment (m_s_) of the samples can be also determined with the SQUID technique, which can be used to determine the iron oxide content of our sample. Therefore, 3 different types of measurements were made: firstly, static hysteresis loop. Secondly, field-cooled (50 Oe = 5 mT) and zero-field-cooled (FC-ZFC) measurements were performed where the magnetic moment was measured of the samples as a function of the temperature. Finally, the saturation magnetic moment (ms) of the samples was determined as a function of the temperature. At low temperature (10 K) with the help of a reference sample of which iron oxide content was known, the iron oxide content of our sample could be determined.

### 2.7. Electron Spin Resonance (ESR) Spectrometer

A commercial ESR spectrometer (Bruker Elexsys, Ettlingen, Germany) was used for the measurements. A microwave irradiation at 9.2 GHz was used as excitation, and the magnetic field was swept between 0 G and 7000 G. In order to determine the magnetic iron oxide content of a lyophilised sample (standard sample: PSI-DAB-Magn), we used a standard, water-based iron oxide suspension as a reference sample from a commercial source (FERROTEC EMG 705). The density of the reference sample was ρ = 1.19 g/mL, and its iron oxide volume fraction was 3.6%, leading to a 0.226 g/mL iron oxide concentration [12].

### 2.8. Magnetic Hyperthermia Measurements

The heating efficiency of the samples in alternating current (AC) calorimetry was tested in a magneTherm™ (Nanotherics, Warrington UK) system [15,19]. The iron oxide content of each sample was measured using the spectrophotometric determination of iron according to the 1,10-phenantroline method described by Mykhalyk et al. [21]. During the magnetic hyperthermic measurements, iron oxide-containing fibrous samples were placed in 1 mL UP water, and then, the heating efficiency was measured with a home-built resonator setup, which generated an intensive alternating magnetic field. The resonator consisted of a 17-turn solenoid coil and a 198 nF capacitor, which yielded a resonance frequency of 109.4 kHz. We employed a 300 s long irradiation with a magnetic field of B = 20.56 mT (H = 16.6 kA/m). Specific absorption rate (SAR) values were calculated from the initial slope of heating curves using the equation:(1)$$SAR=\frac{C_{p,s}}{c}\frac{\Delta T}{\Delta t}(=\frac{C_{p,s}=\sum_{i}C_{pi}m_{i}}{m_{Fe}}\frac{\Delta T}{\Delta t}),$$
where $C_{p,s}=\sum_{i}C_{pi}m_{i}$ is the specific heat of the sample with the unit of J/K ($C_{p,i}$ is the specific heat capacity of the i-th component in the sample), *ρ* is the density of the sample (in the unit of kg/m^3^), “c” is the concentration of the magnetic iron oxide in the sample (kg/m^3^), and ΔT/Δt is the rate of temperature change (K/s) measured experimentally [17]. As a result, Equation (1) yields the unit of J/kg for the SAR, as required. [15].

### 2.9. In Vivo Experiments

In vivo experiments were performed to investigate the in vivo biological response to the PSI-DAB-Magn meshes and their MRI contrast agent efficiency. Three inbred Wistar rats weighing 240–260 g were randomly chosen for this purpose. Animals were bred under specific pathogen-free conditions, kept under standard laboratory conditions (temperature of 20–24 °C, relative humidity of 50–60% and 12 h light/ 12 h dark), fed with a laboratory diet and given water ad libitum. Samples were sterilized prior to implantation with a 30 min immersion in a chlorine dioxide (300 ppm)/physiological saline (0.9 *w*/*w* %) solution [22]. The surgical procedure can be summarized in five steps: (1) general anaesthesia was conducted via the intraperitoneal administration of a Ketamine (70 mg/bodyweight kg, Calypsol 50 mg/mL injected) and Xylazine (10 mg/bodyweight kg, CP-Xylazine 2% injected AUV) (Richter Gedeon Ltd., Budapest, Hungary) mixture in a 4:1 (*V*:*V*) ratio; (2) a 3 cm long incision was made along the median line of the nuchal area (nape); (3) the underlying subcutaneous tissue was dissected and prepared; (4) the PSI-DAB-Magn membranes were fixated along the paramedian line on the underlying muscle tissue with a single simple interrupted suture using atraumatic surgical threads (3/0 polyglycolic acid absorbable suture material; Atramat^®^, International Farmaceutica, Mexico City, Mexico); (5) skin closure was performed with 3 simple interrupted sutures using 2/0 polyglycolic acid absorbable surgical material (Atramat^®^).

Postoperatively, the animals were kept in individual cages and observed daily for evidence of wound complications (inflammation, seroma, etc.). The animals were terminated, and the PSI-DAB-Magn membranes were removed on the 8th postoperative day to determine any macroscopical or microscopical changes in the surrounding area.

The samples were preserved in formaldehyde and subsequently sent for histological evaluation. Tissue samples were fixated in 10% buffered formalin and embedded in paraffin. Sections with a thickness of 4 μm were prepared and then dyed with Haematoxylin-Eosin. To visualise the iron content of the area surrounding the implant, Berlin Blue staining was also used. Glass slides were digitalised with a Panoramic 250 Flash Scanner (3DHISTECH Ltd., Budapest, Hungary).

The experimental protocol adhered to rules laid down by the Directive of the European Parliament and of the Council on the protection of animals used for scientific purposes and was approved by the Semmelweis University’s Institutional Animal Care and Use Committee. The accreditation number of the laboratory is 22.1/1244/3/2015. Control animals were not added to the experiments to reduce the number of involved animals according to 2010/63/EU guidelines of the European Union. Furthermore, the control surgical procedures involving a simple incision, suture fixation on the muscle and skin closure were regarded unnecessary, as physiological wound healing processes are well documented in the literature and pathology bibliography.

### 2.10. MRI

On the 1st and 7th postoperative days, the animals were placed in a nanoScan PET/MR (Mediso, Budapest, Hungary) instrument equipped with a 1 T permanent field magnet, a 450 mT/m gradient system and a volume transmit/receive coil with a diameter of 60 mm. The experimental setup was the following: axial orientation, 0.5 mm slice thickness, 128 number of slices and a pixel size of 0.5 mm. T2 relaxation was determined in 3 animals parallel.

## 3. Results and Discussion

PSI-based fibrous meshes were successfully fabricated by electrospinning similarly to our previous publications [7,13,17]. Electrospun membranes were subsequently treated with a DAB/EtOH solution to create crosslinks between PSI polymer chains (Figure 1, Appendix A). This is a necessary step, as PSI would rapidly hydrolyse to PASP and dissolve under physiological conditions [13].

Imide rings in polymers can react with multivalent amines to form crosslinks, which many researchers have utilized in the past [23,24]. In this specific case, to prove the crosslinking reaction did indeed occur, the PSI and PSI–DAB samples were placed in a PBS for a week (pH: 7.4; ionic strength: 150 mM). While the non-crosslinked PSI dissolved under these conditions [13], the PSI–DAB samples did not due to the created crosslinks. Crosslinking times of 1 h and 2 h were attempted (Appendix A). By changing the time PSI meshes was in the DAB solution, the crosslinker density inside the fibres and between the polymer chains could be influenced. However, the crosslinking density could not be directly and quantitatively determined. After the crosslinking reaction, fibres behaved as gel-fibres, and thus, the surrounding solution could penetrate them without dissolving the polymer. Such gel-fibres could swell and shrink in different solvents or in solvents with different ionic strengths. The more crosslinks there were in a hydrogel, the less it changed its size due to external stimuli [25]. This phenomenon was followed by measuring the diameter of the cylinders at each treatment step (Appendix A).

Due to their gel structure, small molecules could diffuse freely into the fibres and not just between the fibres (Appendix A). Consequently, after the treatment with NaOH, MIONPs were precipitated between and within PSI fibres (Appendix A). This method is an easy and widespread way of creating MIONPs in the literature [20,26,27]. The treatments resulted in a fibrous scaffold loaded with MNPs, which responded to an external magnetic field, evidently exhibiting the presence of iron oxide (Appendix A).

Additionally, by changing the immersion time crosslinked membranes spent in the iron chloride and NaOH solutions, the amount of iron oxide within and between the gel-fibres could be influenced (further details can be found in Appendix A, i.e., Appendix A). The diameter of samples decreased when putting them into the iron chloride solution (due to the change in the solvent from ethanol to water and to the low pH), which again proved that gel fibres were present. During the treatment with NaOH, the diameter of the samples increased because of the high pH value of the solution (Appendix A). The amount of iron oxide present in each sample was determined quantitatively by the 1,10-phenantroline method described by Myhaylyk et al. [21].

SEM studies proved that fibres were not damaged and kept their initial structure during the treatments (Figure 1a,b). It is important to note that fibres did not merge, which is usually the biggest issue when it comes to treating fibrous structures with solutions [28,29]. The average diameter of fibres was 0.87 ± 0.21 μm, which indicated a great variety in diameter, but it is in accordance with other results found in the literature [30]. Most cells favour broader fibre diameter distributions, as such distributions are closer to the structure of the native extracellular matrix [28,29]. The composition of crystals on the surface of fibres seen in Figure 1b was investigated by EDS, which indicated high proportions of iron and oxygen (Figure 1c,d). This proved that iron oxide did not only attach to the surface, but was also present inside the fibres.

The chemical analysis of samples was performed by ATR-FTIR spectroscopy (Figure 1e and Appendix A). On the FTIR spectrum of PSI fibres, the peak at 1710 cm^−1^ indicated the asymmetric stretching vibration of the C=O group, the peak at 1391 cm^−1^ shows the C–O bending vibration, and the peak at 1355 cm^−1^ indicated the C–N stretch of the imide ring [14]. These peaks, however, disappeared on the spectrum of the PSI-DAB-Magn fibrous sample, since all imide rings were crosslinked or opened during NaOH treatment. This sample showed an absorption at 580 cm^−1^ (iron oxide Fe–O stretch), which proved the presence of iron oxide in the membranes [31], as pure iron oxide had the absorption peak at the same position (Figure 1e gray line). Additional spectra can be found in Appendix A.

A small amount of the iron oxide nanoparticles was washed out from the crosslinked fibrous membranes to prove the success of the synthesis and investigate the shape of the particles. TEM studies confirmed that iron oxide was present in the sample as individual nanoparticles rather than as aggregates (Figure 1f and Appendix A). Particles sizes (10 ± 2 nm based on TEM and 25 ± 7 nm based on the DLS measurements) were in the range of what this synthesis method usually produces [15,20].

The ESR studies documented well the magnetic behaviour of the nanoparticles present in the membranes (Figure 2 and Appendix A). The ESR signal of the sample followed the reference signal, which therefore confirmed that the nanoparticles were indeed iron oxide (Figure 2b).

Furthermore, the SQUID studies of the scaffolds at room temperature showed no hysteresis in the investigated range (Figure 2a), which is consistent with the data found in the literature (Nkurikiyimfura et al. found no hysteresis in the SQUID spectrum of iron oxide synthesized with the coprecipitation method at 300 K) [32]. It is worth noting that one sometimes detects a small, about 5 Oe hysteresis or coercivity which is mostly instrumental given the presence of trapped superconducting flux lines in the magnet system. The fact that no coercivity was detected in our samples is a strong proof that the nanoparticles were superparamagnetic. Nanoparticles with a size smaller than the “superparamagnetic diameter”, which is about 70 nm for iron oxide [33], remain superparamagnetic, which is characterised by the absence of coercivity, and are also better suited for radiofrequency hyperthermia [34,35,36]. The maximum magnetization of the PSI-DAB-Magn sample (44 emu/g_iron oxide_) was in the range that is described in the literature. Chen et al. found 57 emu/g as the magnetization for pure MNPs and 2 emu/g for their magnetic fibre [30]. The magnetization of the coated iron oxide at 300 K synthesised by the co-precipitation method is described by Nkurikiyimfura et al. [32], who found this to be 60 emu/g. Our sample showed a magnetization of 44 emu/g at 300 K (Figure 2a), which is acceptable based on the literature, since the polymer fibres in the PSI-DAB-Magn sample did not contribute to magnetization.

The magnetic hyperthermic effect of the PSI-DAB-Magn sample was investigated in a range of frequencies and magnetic fields (Appendix A). Based on the pilot experiments (not shown), we chose 109.4 kHz and 20.56 mT as most effective (Figure 2c,d and Appendix A). The minimum accepted heating rate for an effective hyperthermic treatment accepted by the community is at least a 5 °C temperature increase under 5 min [37,38]. From an initial average 37 °C human body temperature, an increase of 5 °C leads to 42 °C, which is enough to denature proteins in cells and therefore kill them (by inducing cell apoptosis) [39]. The PSI-DAB-Magn samples were able to heat their surrounding by 5 °C in less than 2 min and reach above 10 °C in less than 5 min with an SAR value of 10.8 ± 1.6 W/g. Thus, these samples performed much better than the requirement; therefore, they can be utilised for effective hyperthermic treatments. In the heating–cooling cyclic measurements, the PSI-DAB-Magn samples were able to heat up by at least 8 °C for 6 min in every cycle (Figure 2c). A comparable measurement was carried out by Ghavaminejad et al. [38]. Similar to their results, we also found a uniform cyclic profile, as the temperature of the sample increased by the same amount (around 8 °C) every cycle. This means, after implanting the sample, multiple treatments could be performed with the same efficiency.

The synthesis of magnetic iron oxide in nanofibers has the added benefit to hinder the Brownian relaxation contribution [34,35,36]. The Brownian relaxation is related to the motion of the whole MNP with respect to its viscous surroundings. However, for MNPs with nominal sizes below about 12–15 nm, the so-called Néel relaxation (i.e., when the superparamagnetic magnetization rotates with respect to the particle) dominates the radiofrequency absorption. Thus, the presence of the nanofibers essentially did not influence the heating efficiency.

Cancer cells, however, can become resistant to heat shocks. Thus, it is also important to investigate if samples can induce a hyperthermic effect on a daily basis. Therefore, the hyperthermic effects of the PSI-DAB-Magn samples were investigated every day for five days. During the 5-day repetitive measurement, the samples consistently heated their surrounding by more than 12 °C for 5 min every day (Figure 2d). The respective SAR values calculated on each day are as follows: 12.26 W/g, 11.96 W/g, 10.99 W/g, 8.16 W/g and 10.86 W/g; average: 10.84 ± 1.62 W/g. These cyclic temperature changes are consistent with the data found in the literature [40]. Therefore, the PSI-DAB-Magn samples could be utilised for hyperthermic treatments even days after implantation.

To investigate the biocompatibility of the membranes as well as their potential use as MRI contrast agents, in vivo experiments were performed on Wistar rats (Figure 3a).

There were no difficulties or complications during the surgical procedure. Membranes could be easily handled even with traumatic surgical instruments, while suturing and surgical fixation were easily performed. The mechanical performance of theses meshes is a delicate problem, as the nanoparticle distribution is impossible to regulate the quantification of its effect on the mechanical performance. Nevertheless, the mechanical performance of PSI meshes have been investigated in our previous work [7].

Throughout the entire postoperative period, no complications were macroscopically observed in the surgical area. The animals did not exhibit signs of postoperative pain or irritation, and their behaviours were comparable (food intake, bowel movement and mobility) to those of the control animals kept under the same housing conditions. According to the EU regulation, animal experiments should be minimized, and thus, the control animals for PSI and PSI–DAB meshes were not conducted as cyto- and biocompatibility of these systems presented in a previous work [7].

It is known that MIONPs can effectively be used as contrast agents in MRI [41]. Our goal was to determine whether superparamagnetic MNPs entrapped within an artificial membrane can enhance the contrast in MRI for several days. In the images taken on the 1st post-operative day (Figure 3d), the samples exhibited an excellent contrast capability. The samples were seen with clear dimensions and borders without observable inflammation surrounding them. This is a huge advantage, as these membranes can be used to carefully trace cancerous lesions and then provide information regarding the tumour size changes. Additionally, information about the current iron oxide content of the scaffold can also be obtained, which helps decide whether an additional hyperthermic treatment is a feasible era of new implants needed. The MRI images taken on the 8th day following implantation showed an excellent contrast, thus suggesting that even inside a living organism, the samples did not lose a considerable amount of iron oxide (Figure 3e). Sample dimensions did not change during the 8-day period, and no lesions were observed in the surrounding tissue.

The animals were terminated after eight days (Figure 3b,c). No inflammatory signs or other complications were macroscopically visible neither on the meshes nor in the surrounding tissue. The samples were carefully removed and dissected to retain the samples intact while preserving the surrounding newly formed granulation tissue with the underlying muscle layer.

Histological sections were stained with haematoxylin and eosin to assess the inflammatory and other tissue reactions or Berlin Blue to examine the release distribution of the iron MNPs. Histopathology reviled a mild inflammatory reaction (Figure 4a,b). However, an inflammatory reaction is expected in every type of surgical procedure, particularly when the procedure involves the implantation of foreign objects [42,43]. Nevertheless, no serious complications were observed. The granulation tissue was rather thin, while cells had already started to infiltrate the membrane. In other words, the scaffold may provide a template that cells can adhere to and proliferate on. This is especially important, as the tissue regenerates after cell death. One week was certainly too short to draw a decisive conclusion, but the preliminary results were very promising (minimal inflammation and no foreign body type giant cells) and suggested that the samples were indeed biocompatible.

Berlin blue staining was very efficient in visualizing the iron nanoparticles (Figure 4c,d). There was a clearly visible border between the membrane and the local granulation tissue [43]. However, traces of iron nanoparticles could be found further away from the sample, which suggested that tissue integration was in progress.

In vivo results suggested that these meshes were not only biocompatible, but also functional. They withstood surgical manipulation and fixation and also induced only minimal inflammation while slowly releasing their iron oxide content. Although biodegradation was not visible, we expect these samples to degrade within a month. This system with somewhat fast biodegradation time can be an advantage compared to other polymer systems, as those are prone to induce a stronger response, sometimes even chronic inflammation and thus more complications [44]. However, this has yet to be tested. Therefore, in the next step, we would like to observe the long-term effect, biodegradation and the decay of the hypothermic effect in vivo.

## 4. Conclusions

To conclude, a fibrous PSI scaffold loaded with iron oxide nanoparticles was fabricated for theranostic application. Polymer chains inside the fibres were crosslinked with DAB, creating a fibrous hydrogel structure. Iron oxide nanoparticles were synthesised inside and between the fibres using the co-precipitation method. PSI-DAB-Magn samples kept their extracellular matrix-like structures after treatments, and the fibres did not merge. During magnetic hyperthermic measurements, the samples repeatedly increased in temperature more than 8 °C for 5 min, which was above the minimum criteria of 5 °C for 5 min described in the literature. Therefore, the PSI-DAB-Magn samples could successfully induce local hyperthermia. The MRI studies confirmed that these samples could be used as contrast agents, as they gave good contrast on the 1st and 7th days following implantation. The histopathologic evaluation of the PSI-DAB-Magn samples showed a minimal acute inflammation on the 8th day following implantation, which meant the samples were likely to be biocompatible. Furthermore, staining the sections with berlin blue confirmed that iron oxide not only was present at the site of the implantation even after eight days, but did not diffuse either. Therefore, a localised hyperthermic treatment is feasible. Therefore, PSI-DAB-Magn membranes seem to be an excellent candidate for the potential supplementary management of cancer.

## Data Availability

Not applicable.

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
