# Peer review of "An Implantable Magneto-Responsive Poly(aspartamide) Based Electrospun Scaffold for Hyperthermia Treatment"

_nanomaterials, 2022, doi:10.3390/nano12091476_

Round 1

Reviewer 1 Report

The manuscript “An implantable magneto-responsive poly(aspartamide) based electrospun scaffold for hyperthermia treatment” by Veres and Voniatis et al. is a very well-written manuscript which the synthesis and characterization of an electrospun scaffold containing magnetic nanoparticles for the use in hyperthermia treatments.

The manuscript is well written. For the references, the issue arises that one author (the correspondng author) is cited 15 times with other co-autors also being cited in different publications, there is a slight imbalance in the references. There should be less than a fourth or a fifth of self-publications in a manuscript, otherwise, there is no real objectivity in the discussion and the state of the art. The authors should reduce the self-citations significantly.

Moreover, there are multiple reference errors all over the manuscript which need to be fixed.

Other than that, the methods section is well elaborated and the manuscript outline is well written and the results are well discussed.

I am not happy about the wording magnetite nanoparticles since you do not show any differentiation to any other iron oxides such as maghemite particles. It is quite difficult to very well differentiate between iron oxides on the nanoscale level with the surface being very prone to oxidation. I also do not think that you need to verify the magnetite content with multiple techniqus (e.g. Mossbauer spectroscopy) but you should use a broader term instead of using just magnetite or pure magnetite as stated in the following sentence: “This sample showed an absorption at 580 cm-1 (magnetite Fe-O stretch), which proves the presence of magnetite in the membranes [35], as pure magnetite has the absorption peak at the same position (Figure 1 (e) orange line).”

Maghemite also shows a Fe-O stretch vibration at 580 cm-1 and even shows the same shoulder as your “pure magnetite” sample. Maybe a less absolute term such as iron oxide or magnetic iron oxide is more suitable here.

Figure 2 a: The y-axis shoxs emu in g but should actually represent the specific electromagnetic units (emu/g)

Figure S3 is not referenced in the manuscript

On page 16 the iron nanoparticles (should be iron oxide nanoparticles).

I would recommend minor revisions since I very much like the outline and the methodology used in this manuscript.

Author Response

Reviewer 1:

The manuscript “An implantable magneto-responsive poly(aspartamide) based electrospun scaffold for hyperthermia treatment” by Veres and Voniatis et al. is a very well-written manuscript which the synthesis and characterization of an electrospun scaffold containing magnetic nanoparticles for the use in hyperthermia treatments.

1, The manuscript is well written. For the references, the issue arises that one author (the correspondng author) is cited 15 times with other co-autors also being cited in different publications, there is a slight imbalance in the references. There should be less than a fourth or a fifth of self-publications in a manuscript, otherwise, there is no real objectivity in the discussion and the state of the art. The authors should reduce the self-citations significantly.

The issues with the references was corrected, 9 self-citations were deleted from the Reference list.

2, Moreover, there are multiple reference errors all over the manuscript which need to be fixed.

We do not understand this issue, maybe during the pdf conversion probably a technical error occurred because in the word file what we uploaded did not find anything like that just on the conversed file, thus now we are double checking that conversion. Thank you very much for this notice.

3, Other than that, the methods section is well elaborated and the manuscript outline is well written and the results are well discussed.

Thank you very much for this comment.

4, I am not happy about the wording magnetite nanoparticles since you do not show any differentiation to any other iron oxides such as maghemite particles. It is quite difficult to very well differentiate between iron oxides on the nanoscale level with the surface being very prone to oxidation. I also do not think that you need to verify the magnetite content with multiple techniqus (e.g. Mossbauer spectroscopy) but you should use a broader term instead of using just magnetite or pure magnetite as stated in the following sentence: “This sample showed an absorption at 580 cm-1 (magnetite Fe-O stretch), which proves the presence of magnetite in the membranes [35], as pure magnetite has the absorption peak at the same position (Figure 1 (e) orange line).”

Maghemite also shows a Fe-O stretch vibration at 580 cm-1 and even shows the same shoulder as your “pure magnetite” sample. Maybe a less absolute term such as iron oxide or magnetic iron oxide is more suitable here.

Thank you very much for this important comment, you are absolutely right, we did not perform any measurements to proof the magnetite and did not use any coverage, thus in the manuscript the magnetite nanoparticles, were changed for “magnetic iron oxide nanoparticles (MIONs)”.

5, Figure 2 a: The y-axis shoxs emu in g but should actually represent the specific electromagnetic units (emu/g)

The y axis was corrected in the following way:

Specific magnetization (emu/g)

6, Figure S3 is not referenced in the manuscript

Thank you very much for the comment, now the figure reference was placed on the manuscript:

“Additionally, by changing the immersion time crosslinked membranes spend in the iron-chloride and NaOH solutions, the amount of iron oxide within and between the gel-fibres can be influenced (further details can be found in the supplementary information, Figure S3).”

7, On page 16 the iron nanoparticles (should be iron oxide nanoparticles).

Thank you very much for the particular section was in the manuscript was revised.

I would recommend minor revisions since I very much like the outline and the methodology used in this manuscript.

Reviewer 2 Report

The authors and colleagues report an implantable electrospun scaffold containing Polysuccinimide (PSI), 1,4-diaminobutane (DAB) and Fe(II)-Fe(III)-chloride. However, major revision should be done to further improve the quality of this manuscript. Samples could induce a magnetic hyperthermia of 5 ℃ (from 37 ℃ to 42 ℃) in under 2 minutes, at the same time, it has good biocompatibility, degradability and MRI imaging. However, major revision should be done to further improve the quality of this manuscript.

  1. In vitro experiments are not presented throughout the text, and cell experiments on photothermal properties of electrospum scaffolds need to be reflected in the manuscript.
  2. XRD, XPS, HRTEM, Mapping of electrospum scaffolds should be provided.
  3. Are there longer experiments to verify the biodegradability of the material and whether the amount of degradation can be semi-quantified?
  4. The rheological property data of the samples should be mentioned in the manuscript.
  5. Some relevant work should be cited (Chemical Society Reviews, 2021, 50(15): 8669-8742; Chemical Science, 2019, 10(21): 5435-5443).

Author Response

Reviewer 2:
The authors and colleagues report an implantable electrospun scaffold containing Polysuccinimide (PSI), 1,4-diaminobutane (DAB) and Fe(II)-Fe(III)-chloride. However, major revision should be done to further improve the quality of this manuscript. Samples could induce a magnetic hyperthermia of 5 ℃ (from 37 ℃ to 42 ℃) in under 2 minutes, at the same time, it has good biocompatibility, degradability and MRI imaging. However, major revision should be done to further improve the quality of this manuscript.
1, In vitro experiments are not presented throughout the text, and cell experiments on photothermal properties of electrospum scaffolds need to be reflected in the manuscript.
We had many issues with the in vitro experiments. The biggest problem is that the magnetic nanoparticles absorb the light, thus all cytotoxicity measurements we could perform in our lab (MTT assy, WST1 measurements) is useless in that case due to the interference of the light for the concentration determination. We know from our previous studies that the magnetic nanoparticles are not toxic (Int. J. Mol. Sci. 2013, 14, 14550-14574; doi:10.3390/ijms140714550, http://dx.doi.org/10.1016/j.colsurfa.2014.01.043) and hemocompatibly and we have the same knowledge for the scaffold (https://doi.org/10.1371/journal.pone.0254843 and another paper is under submission, waiting for the acceptance after 2 minor reviewer comment).
So please find below the scaffold cytotoxicity details with healthy fibroblast cells which is under submission at this point:
But in the presence of the scaffold filled with magnetic particles is still an issue, and we are currently working towards a solution.
2, XRD, XPS, HRTEM, Mapping of electrospum scaffolds should be provided.
The reviewer is right, but these measurements will be part of another manuscript. We are dealing with the best sample preparation for these techniques. Our samples are fabricated in a wet form after the chemical crosslink and the nanoparticle synthesis. The samples were used in that form for the experiments. After freeze drying the structure is changing and of course we can get information about the crystalline phase of the iron oxide and we can get some information from the HR-TEM also, but in this paper our aim was to get an implantable scaffold for ongoing treatment. During the freeze drying the scaffold became rigid and fragile which is not in the case of wet form. Because we did not use any protection layer at this time for the iron oxide, based on our previous experience the nanoparticles have a magnetite core and a maghemite oxidized layer outside (a core shell structure). Because the magnetic behavior is not changing that much with the oxidation (showing in the manuscript on the Figure 2), the specific magnetization of the scaffold is 44 emu/g, which is not that much lower than the ferrofluids (between 50-80 emu/g based on the literature), we did not perform that much characterization on the scaffold.
For the particles the following XRD can be find:
3, Are there longer experiments to verify the biodegradability of the material and whether the amount of degradation can be semi-quantified?
That was an 8 days experiment and we plan to focus on a longer experiments with animals 1,2,36 months long for biodegradation measurements. In this case the macroscopic structure and the microscopic analyzation during the histopathology can give us an answer for the degradation phoneme. Thus that kind of long experiment will be part of another manuscript in the future.
4, The rheological property data of the samples should be mentioned in the manuscript.
For the authors the rheological property data is not exactly clear how to perform. We have a fibrous scaffold filled with nanoparticles and water/buffer. In the case of electrospun fibrous scaffold a proper rheological measurement cannot be done. The magnetic particles were synthetized inside the scaffold, thus comparing a magnetic fluid with the scaffold is not relevant in our system. And placing the sample in a rheometer not sure what kind of information we can get from it.
Rheology can be done on the polymer solution before the electrospinning, but it can give information for the fiber formation and it is not relevant for the final product which goes through chemical modification after the fiber formation and wet chemistry for the nanoparticle formation. Thus we can
not do that kind of measurements on the final product which was placed in the hyperthermic instrument or implanted to the animals.
5, Some relevant work should be cited (Chemical Society Reviews, 2021, 50(15): 8669-8742; Chemical Science, 2019, 10(21): 5435-5443).
Thank you very much for your suggestion. The first paper is very relevant and we will cite it, although the second one is showing results about pegylated rhenium nanoclusters and there photothermal therapy, which we believe is out of scope.

Reviewer 3 Report

Comments to the Authors:

The submitted paper describes a magneto-responsive fibrous scaffolds that could be potentially used in magnetic hyperthermia therapy. The Authors performed numbers of experiments to characterize their magnetic materials including in vivo studies. However, the MAJOR REVISION of the manuscript is required.

  1. The style of English is not equal in the whole manuscript and has to be corrected as some of the sentences makes the description of the experiments totally not clear for the readers. The manuscript and supplementary materials can be accepted for publication ONLY after the careful proofreading of the text, preferably by a native speaker or a professional proof-reading service. Generally, the layout of the manuscript and its technical side also leaves much to be acceptable.
  2. Apart from the numerous typos, grammar misuses and technical problems, some of the sentences are not understandable. For instance, in the Abstract one can read: “To localize the nanoparticles they were synthesized inside fibres.” What does it mean to localize particles (where and for what?)
  3. Then, the Authors said that “Samples could induce a magnetic hyperthermia of 5 ℃…” In my opinion, the magnetic hyperthermia is the process of heating or the therapeutic approach. The temperature increase is the easiest predictor to measure the efficiency of this process but the hyperthermia is not equal to the temperature increase.
  4. “Amongst many, nanoparticles have the potential to be utilised as theranostics agents” In this sentence, it is not clear what we are referred to (among many what?).
  5. The Authors used “theranostic agents” and “theranostics” as synonyms. For readers familiar with this issues, it is clear that the theranostics is the paradigm and agents “only” can make this paradigm work. Speaking of which, in the 2. Paragraph in the Introduction section the references about theranostics are missing. Recently, numbers of review papers were published in this topic, e.g. https://www.thno.org/v11p10091 or https://www.thno.org/v11p8706.htm and many others.
  6. The Authors have claimed that “Magnetic hyperthermia has been used as a complementary treatment for cancer be-sides chemotherapy and radiotherapy aiming to reduce the amount of medications and radiation needed, consequently decreasing their side-effects as well”. In many works, magnetic hyperthermia has been described as a promising method, however its practical usage is limited until now. The honest and specific explanation about that would be necessary.
  7. In Materials and Methods section (2.1.) the first sentence is not a sentence. It should be checked.
  8. Then, the Authors used the expression: “the in situ synthesis of magnetite NPs” . It is not clear, however, if it is a novel contribution of current paper or it has been introduced by others. The corresponding explanation and/or references are missing.
  9. The description of the magnetic hyperthermia setup is quite vague: “the heating effi-ciency was measured at 109.4 kHz for 300 s with a magnetic field of B = 20.56 mT (H = 16.6 kA/m, 17 turn coil and 198 nF capacitor).” It should be more specific and linguistically improved. The same concerns the Equation 1 that is not properly edited (parenthesis) and its description is inconsistent as some of the parameters are in italics, another not.
  10. What is the unit of SAR values resulting from the Equation 1?
  11. The Authors used the acronym SPMNPs for their nanoparticles. In the literature, one can meet rather It might help the readers in searching for this paper to use the well-known terminology, although it is not mandatory.
  12. In the whole text of the manuscript, one can meet the text: Error! Reference source not found. It has to be fixed.
  13. Schematic 1 seems to not be mentioned in the text of manuscript. It must be checked. Also, the caption of Schematic 1 is poor. It should be self-explaining for readers that are interested in the general highlights of the work. For instance, it is not clear what the objects on the left side are: the gel phantoms or something similar to them? It should be included in the caption.
  14. Additionally, by changing the immersion time crosslinked membranes spend in iron-chloride and NaOH, the amount of magnetite within and between the gel-fibres can be influenced (further details can be found in the supplementary information). “ There should be specific reference to the given part of supplementary materials (figure or table).
  15. The Authors investigated magnetic heating of nanoparticles entrapped into gel-like structure. The advantages of such solution are stressed out in the manuscript in a convincing way. However, it is well-known that from fundamental point of view, heating efficiency of magnetic heating for superparamagnetic particles depends on the efficiency of Neél and Brownian relaxation and the latter one can be inhibited for very viscous and gel-like media (see e.g. https://ujp.bitp.kiev.ua/index.php/ujp/article/view/2020073 and https://www.mdpi.com/2079-4991/9/5/803 ). The discussion of the results from calorimetric studies should be expanded and the efficiency of heating should be discussed. It can be also the suggestion for the further studies where such scaffolds could be compared to magnetic fluids.
  16. What is more, the text of the manuscript should be checked again due to the technical requirements of the Journal. For instance, the references to the Figures and Equations in the text of the manuscript or the format of equations. The small inconsistences also appear in the layout of the Tables.

Author Response

Reviewer 3:
The submitted paper describes a magneto-responsive fibrous scaffolds that could be potentially used in magnetic hyperthermia therapy. The Authors performed numbers of experiments to characterize their magnetic materials including in vivo studies. However, the MAJOR REVISION of the manuscript is required.
Comments:
1, The style of English is not equal in the whole manuscript and has to be corrected as some of the sentences makes the description of the experiments totally not clear for the readers. The manuscript and supplementary materials can be accepted for publication ONLY after the careful proofreading of the text, preferably by a native speaker or a professional proof-reading service. Generally, the layout of the manuscript and its technical side also leaves much to be acceptable.
Thank you very much for the comment, the entire manuscript was revised and resubmitted.
2, Apart from the numerous typos, grammar misuses and technical problems, some of the sentences are not understandable. For instance, in the Abstract one can read: “To localize the nanoparticles they were synthesized inside fibres.” What does it mean to localize particles (where and for what?)
Thanks for this comment it was changed for “to immobilize”. All the typos and grammar mistakes were fixed in the manuscript.
“To immobilize the nanoparticles inside the scaffold, they were synthesized inside hydrogel fibres.”
3, Then, the Authors said that “Samples could induce a magnetic hyperthermia of 5 ℃…” In my opinion, the magnetic hyperthermia is the process of heating or the therapeutic approach. The temperature increase is the easiest predictor to measure the efficiency of this process but the hyperthermia is not equal to the temperature increase.
The sentence was corrected in the following way:
“The magnetic hyperthermia efficiency of the samples was significant. The given AC magnetic field could induce a temperature rise of 5 ℃ (from 37 ℃ to 42 ℃) in under 2 minutes even for 5 quick heat-cool cycles or for 5 consecutive days without considerable heat generation loss in the samples.”
4, “Amongst many, nanoparticles have the potential to be utilised as theranostics agents” In this sentence, it is not clear what we are referred to (among many what?).
The mentioned sentence was corrected in the following way:
“Amongst many possible applications, nanoparticles have the potential to be utilized as theranostics agents.”
5, The Authors used “theranostic agents” and “theranostics” as synonyms. For readers familiar with this issues, it is clear that the theranostics is the paradigm and agents “only” can make this paradigm work. Speaking of which, in the 2. Paragraph in the Introduction section the references about theranostics are missing. Recently, numbers of review papers were published in this topic, e.g. https://www.thno.org/v11p10091 or https://www.thno.org/v11p8706.htm and many others.
Thank you very much, now it is corrected and one of the paper is cited:
“Theranostic agents (used in theranostics) concurrently combine both therapeutic and diagnostic features, a feat particularly advantageous in cancer management. These ensure that no undesirable
differences occur between the biodistribution of diagnostic and therapeutic agents. The long-term goal of using theranostic agents is to monitor the effects of treatments while fine-tuning the therapy according to the specific needs of the patient, thus achieving personalized medicine [5].”
6, The Authors have claimed that “Magnetic hyperthermia has been used as a complementary treatment for cancer be-sides chemotherapy and radiotherapy aiming to reduce the amount of medications and radiation needed, consequently decreasing their side-effects as well”. In many works, magnetic hyperthermia has been described as a promising method, however its practical usage is limited until now. The honest and specific explanation about that would be necessary.
Thank you very much for this notice, now the sentence was corrected in the following way (and a new citation was added):
“Magnetic hyperthermia is still a promising complementary treatment (its in preclinical phase) for cancers besides chemotherapy and radiotherapy aiming to reduce the amount of medications and radiation needed, consequently decreasing their side-effects as well [145].”
7, In Materials and Methods section (2.1.) the first sentence is not a sentence. It should be checked.
Now it is corrected.
8, Then, the Authors used the expression: “the in situ synthesis of magnetite NPs” . It is not clear, however, if it is a novel contribution of current paper or it has been introduced by others. The corresponding explanation and/or references are missing.
Thanks for the comment “in situ” was deleted from the relevant part to avoid the misunderstanding.
9, The description of the magnetic hyperthermia setup is quite vague: “the heating effi-ciency was measured at 109.4 kHz for 300 s with a magnetic field of B = 20.56 mT (H = 16.6 kA/m, 17 turn coil and 198 nF capacitor).” It should be more specific and linguistically improved. The same concerns the Equation 1 that is not properly edited (parenthesis) and its description is inconsistent as some of the parameters are in italics, another not.
Thank you very much for this comment now the manuscript was corrected in the following way:
"then the heating efficiency was measured with a home-built resonator setup which generates an intensive alternating magnetic field. The resonator consists of a 17 turn solenoid coil and a 198 nF capacitor, which yields a resonance frequency of 109.4 kHz. We employed a 300 s long irradiation with a magnetic field of B = 20.56 mT (H = 16.6 kA/m)."
10, What is the unit of SAR values resulting from the Equation 1?
Thank you very much for your note, the manuscript was corrected in the following way:
“???=??,??cΔ?Δ?(=Σ?????????Δ?Δ?) (Eq. 1)
where ??,?=Σ?????? is the specific heat of the sample with units of J/K (??,? is the specific heat capacity of the i-th component in the sample), ρ is the density of the sample (in units of kg/m3), "c" is the concentration of the magnetic iron oxide in the sample (kg/m3) and ΔT/Δt is the rate of temperature change (K/s) measured experimentally [17]. As a result, Eq. 1 yields units of J/kg for the SAR, as required. [13].”
11, The Authors used the acronym SPMNPs for their nanoparticles. In the literature, one can meet rather It might help the readers in searching for this paper to use the well-known terminology, although it is not mandatory.
The relevant part was corrected.
12, In the whole text of the manuscript, one can meet the text: Error! Reference source not found. It has to be fixed.
We do not understand this issue, maybe during the pdf conversion something happened, because in the word file that we uploaded we did not experience any issues. Thank you very much for this notice.
13, Schematic 1 seems to not be mentioned in the text of manuscript. It must be checked. Also, the caption of Schematic 1 is poor. It should be self-explaining for readers that are interested in the general highlights of the work. For instance, it is not clear what the objects on the left side are: the gel phantoms or something similar to them? It should be included in the caption.
Schematic 1 was referred on the second sentence of the Results and Discussion:
“Electrospun membranes were subsequently treated with a DAB/EtOH solution to create crosslinks between PSI polymer chains (Hiba! A hivatkozási forrás nem található., Figure S1 (b) and S2 (a)).”
Now the legend was modified in the following way:
“Schematic representation of the chemical treatment of the PSI scaffold. After preparing the fibrous sample, inside the fibers crosslinks were created between the polymer chains with DAB, then merging the system in iron-chloride and NaOH, nanoparticle formation happened.”
14, “Additionally, by changing the immersion time crosslinked membranes spend in iron-chloride and NaOH, the amount of magnetite within and between the gel-fibres can be influenced (further details can be found in the supplementary information). “ There should be specific reference to the given part of supplementary materials (figure or table).
Thank you very much for this comment, now the citation of the table and figures were modified:
“Additionally, by changing the immersion time crosslinked membranes spend in iron-chloride and NaOH, the amount of iron oxide within and between the gel-fibres can be influenced (further details can be found in the supplementary information, Table S1, Figure S1-3).”
15, The Authors investigated magnetic heating of nanoparticles entrapped into gel-like structure. The advantages of such solution are stressed out in the manuscript in a convincing way. However, it is well-known that from fundamental point of view, heating efficiency of magnetic heating for superparamagnetic particles depends on the efficiency of Neél and Brownian relaxation and the latter one can be inhibited for very viscous and gel-like media (see e.g. https://ujp.bitp.kiev.ua/index.php/ujp/article/view/2020073 and https://www.mdpi.com/2079-4991/9/5/803 ). The discussion of the results from calorimetric studies should be expanded and the efficiency of heating should be discussed. It can be also the suggestion for the further studies where such scaffolds could be compared to magnetic fluids.
The reviewer is correct, we added the following section in the discussion regarding the relevance of nanofibers in hindering or prohibiting the Brownian relaxation:
"The synthesis of magnetic iron oxide in nanofibers have the added benefit to hinder the Brownian relaxation contribution [34-36]. The Brownian relaxation is related to the motion of the whole
magnetic nanoparticle with respect to its viscous surroundings. However, for magnetic nanoparticles with nominal sizes below about 12-15 nm, the so-called Néel relaxation (i.e. when the superparamagnetic magnetization rotates with respect to the particle) dominates the radiofrequency absorption, thus the presence of the nanofibers essentially does not influence the heating efficiency."
16, What is more, the text of the manuscript should be checked again due to the technical requirements of the Journal. For instance, the references to the Figures and Equations in the text of the manuscript or the format of equations. The small inconsistences also appear in the layout of the Tables.
The requirements of the Journal are not so strict at all, thus we decided to use an almost uniform style for the figures. The reference list was made with Mendeley used a common style. The reference of Figures now the same and we put 2 equations to the text with reference also. The only table in the SI is quit a difficult question. We tried to summarize all the variations during the optimization and we hope that it is clear to the reader.

Round 2

Reviewer 2 Report

All issues have been addressed and so I recommend its publication in Nanomaterials.

Reviewer 3 Report

The manuscript was corrected according to the remarks. However, some reference codes are still not properly presented.